# Towards Global-level Mechanistic Interpretability: A Perspective of Modular Circuits of Large Language Models

**Yinhan He** [1]  **Wendy Zheng** [2]  **Yushun Dong** [3]  **Yaochen Zhu** [1]  **Chen Chen** [2]  **Jundong Li** [1]

## Abstract

Mechanistic interpretability (MI) research aims to understand large language models (LLMs) by identifying computational circuits—subgraphs of model components with associated functional interpretations (FIs)—that explain specific model behaviors. Current MI research mainly focus on discovering task-specific circuits, which have two key limitations: (1) low generalization ability across diverse language tasks and (2) high costs due to the need for human or advanced LLMs to interpret each computational node. To address these challenges, we propose a novel **modular circuit (MC) vocabulary** of task-agnostic functional units, each containing a small computational subgraph with its interpretation obtained by examining the subgraph's behavior on extensive corpora. By allowing different language tasks to share common MCs, our approach enables global interpretability while reducing costs by reusing established interpretations for new tasks. Besides, we propose five criteria for characterizing the MC vocabulary and present **ModCirc**, a novel global-level MI framework for discovering MC vocabularies in LLMs. We demonstrate ModCirc's effectiveness by showing that it can identify modular circuits that perform well on various metrics. [1]

## 1. Introduction

The rapid advancement of Large Language Models (LLMs) (Achiam et al., 2023; Anthropic, 2024; Team et al.,

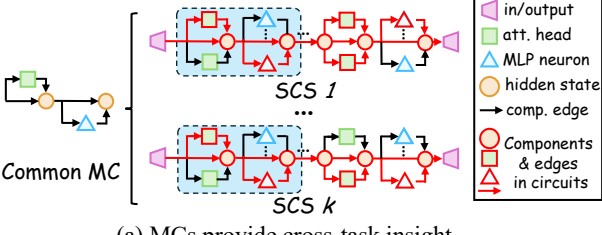

(a) MCs provide cross-task insight.

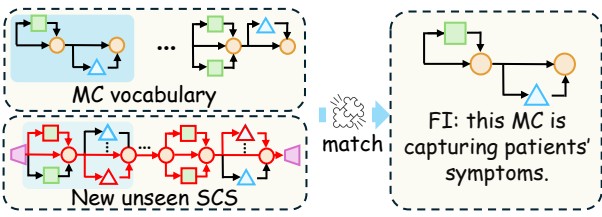

(b) Func. interpretations (FI) are extracted from MC vocab.

*Figure 1.* Advantages of modular circuits (MC) as an LLM mechanistic interpretation tool. (a) Common MCs (shown in sky blue) shared across significant computational subgraphs (SCSs) of different language tasks reveal cross-task interpretability insights. (b) MCs identified in new tasks can be matched against a vocabulary of known MCs, allowing direct reuse of existing functional interpretations (FI) rather than generating new ones.

2023; Chang et al., 2024; Wei & Liu, 2025) has triggered the interest to understand their internal mechanisms and decision-making processes. Mechanistic interpretability (Wang et al., 2022; Lieberum et al., 2023; Cunningham et al., 2023; Nanda et al., 2023; Conmy et al., 2023; Bereska & Gavves, 2024; Rai et al., 2024; Nainani et al., 2024; Nainani, 2024), which aims to reverse engineer how these models process information and make decisions, has emerged as a promising approach. For various language tasks such as indirect object identification (IOI) (Wang et al., 2022), modular sum (Nanda et al., 2023), and subject-verb agreement (Marks et al., 2024), current research has successfully identified many computational circuits, i.e., the significant computational subgraph (SCS) of the LLM that contributes the most to the LLM's output, along with the nodes' functional interpretations (FIs). Specifically, the nodes in these computational subgraphs are LLM components such as attention heads and feed-forward neurons, with edges representing direct computational flow between

---

[1]Department of Electrical and Computer Engineering, University of Virginia, Charlottesville, VA, USA [2]Department of Computer Science, University of Virginia, Charlottesville, VA, USA [3]Department of Computer Science, Florida State University, Tallahassee, FL, USA. Correspondence to: Jundong Li <jl6qk@virginia.edu>.

*Proceedings of the $42^{nd}$ International Conference on Machine Learning*, Vancouver, Canada. PMLR 267, 2025. Copyright 2025 by the author(s).

[1]The code project is available at https://github.com/YinhanHe123/ModCirc.

nodes. For each node, the FI provides textual explanations of its semantic role in completing the language task. Together, the nodes and their interactions form a circuit that explains how an LLM performs a specific language task.

However, existing methods can only identify circuits on a task-by-task basis (i.e., analyzing how models handle each individual task), which presents two key limitations: (1) ***Lack of Generalization ability***: Since the current circuit discovery methods find circuit task by task, they fail to capture LLMs' broader computation patterns shared across different tasks. Hence, the insights gained from studying one task provide little information about how the model approaches other potentially related tasks. (2) ***High Cost***: Current methods generate FI for each node of the identified computational subgraph, which requires costly human inspection (or most advanced LLMs such as GPT-4 (Achiam et al., 2023)) to understand its role. This becomes increasingly impractical as LLMs rapidly scale in size.

To address these limitations, we propose a novel **modular circuits (MC) vocabulary**, as inspired by Nanda (2023)'s hypothesis on the existence of circuit modularity. As shown in Fig. 1, each MC in our vocabulary is a *task-agnostic functional unit consisting of a small computational subgraph (referred to as "modular SCS") within an LLM and its associated FI*. First, it provides global interpretability by enabling different tasks to share common MCs in their circuits. As illustrated in Fig. 1(a), we identify $k$ SCSs across $k$ different tasks. These SCSs share a common modular SCS (highlighted in sky-blue), whose corresponding FI offers cross-task insights. Second, the MC vocabulary reduces the computational cost of circuit discovery, as shown in Fig. 1(b). When analyzing an unseen SCS for a new language task, we can match it against our existing MC vocabulary. This allows us to directly apply the cached FIs from the vocabulary to the matched portions of the SCS, eliminating the need for costly manual inspections.

In our work, we take the first step to find the MC vocabulary in an LLM. We first identify five evaluation criteria (consistency, locality, reusability, composability, and globality) that characterize the properties an ideal MC vocabulary must possess. Based on these evaluation criteria, we subsequently propose a novel **Mod**ular **Circ**uit vocabulary discovery method termed ModCirc. Specifically, with a set of training tasks' SCSs of an LLM, our ModCirc first discovers the reusable computational subgraphs across those circuits according to the reusability criterion. For each of the reusable computational subgraph, we process it with a reinforcement learning-based graph neural partitioning approach to obtain the modular SCSs (each partition in each reusable computational subgraph is considered as a modular SCS). In this process, ModCirc optimizes the consistency, locality, and composability of the partitioning for

each reusable computational subgraph. Finally, we design a novel prompting strategy to generate FIs for the modular SCSs to obtain the MC vocabulary with an advanced LLM.

Our contribution can be concretely summarized as follows: (1) ***Novel formulation***: we formulate the novel problem of MC vocabulary discovery for LLMs. The problem is built upon our designed comprehensive set of evaluation criteria that characterize the quality of an MC vocabulary. (2) ***Algorithm Design***: we propose a principled MC vocabulary discovery framework that provides cross-task interpretability insights and alleviates the financial cost of circuit discovery. (3) ***Experimental Validation***: we conduct extensive evaluations on an LLM across diverse language tasks from multiple domains, which both quantitatively and qualitatively verify ModCirc's effectiveness.

## 2. Preliminaries

Here, we elaborate on how the significant computational subgraph (SCS) of a circuit is computed in existing literature, and the related terminology used throughout our work.

In existing research, the SCS of a circuit on task $T$, denoted as $C_T$, is identified through "activation patching" (Wang et al., 2023; Marks et al., 2024; Zhang & Nanda, 2023; Hanna et al., 2024a; Lieberum et al., 2023) across all computational nodes (abbreviated as "nodes", which refer to LLM components such as attention heads and feed-forward neurons) of the examined LLM. For a node $v$, "activation patching" calculates the its causal indirect effect (IE):

$$\text{IE}(\text{LLM}, v) =$$
$$\mathbb{E}_{d \sim \mathcal{D}} \left| \text{LLM}(d^{clean} | do(v = v^{patch})) - \text{LLM}(d^{clean}) \right|. \quad (1)$$

Here, $\mathcal{D}$ is the input text dataset for task $T$ (e.g., in a disease diagnosis task, the input is a patient's symptom description and the output is the corresponding disease name), and $d = (d^{clean}, d^{patch})$ is a data sample from dataset $\mathcal{D}$. Specifically, $d^{clean}$ is the original input text, while $d^{patch}$ is a corrupted version of the input, typically created by replacing key words in $d^{clean}$ with random words. The term $\text{LLM}(d^{clean})$ represents the logit value from the LLM's output corresponding to the ground-truth answer token (typically a multiple choice letter like "A" or "B"). Meanwhile, $\text{LLM}(d^{clean} | do(v = v^{patch}))$ represents the logit value for the same answer token, but with a key difference: while the input remains $d^{clean}$, the activation (i.e., output) at node $v$ is replaced with $v^{patch}$ (the activation that would have been produced by inputting $d^{patch}$). Nodes with "IE" values higher than a predefined threshold compose the nodes of $C_T$. Two nodes in $C_T$ are connected if there is a direct computation flow between them. Causal tracing will serve as a useful part in our proposed method for finding MC vocabulary $\mathcal{S} = \{S_1, ..., S_k\}$ ($k$ is the number of MCs).

# 3. Evaluation Criteria and Problem Definition

The MC vocabulary is characterized by the following criteria derived from initial discussions of Nanda (2023).

The *first* criterion is **Consistency.** This criterion stipulates that an MC (denoted as $S$) should maintain functional consistency across different tasks. Denote the FI of $S$ when LLM performs task $T \in \mathcal{T}$ (where $\mathcal{T}$ is a set of tasks in consideration) as $FI(S, T)$, the consistency can be evaluated as the maximum number of synonymous FIs of the MC $S$ across tasks $\mathcal{T}$ (denote synonymity by $\equiv$)

$$\textbf{Consist.}_{\mathcal{T}}(S) = \max\{\frac{|\mathcal{M}|}{|\mathcal{T}|} : \mathcal{M} \subseteq \mathcal{T}, \forall T_1, T_2 \in \mathcal{M},$$
$$FI(S, T_1) \equiv FI(S, T_2)\}. \tag{2}$$

The *second* criterion is **Locality**. This criterion mandates that nodes within an MC exhibit both physical and semantic proximity. Physical locality means that nodes must be internally connected and localized with respect to their physical positions within the LLM. Semantic locality indicates that nodes must serve closely related semantic functionalities.

To evaluate locality, we construct two types of embeddings for all nodes in the LLM, a semantic embedding measuring functional behavior similarities between MCs, and a physical embedding measuring structural proximity between the MCs. For semantic embedding, we build a space where each node $v \in \mathcal{V}$ maps to a vector, with distances $d^{sem}(\cdot, \cdot)$ representing functional similarities (see Section 4). For physical embedding, we place nodes in a Euclidean grid based on their proximity (Liu et al., 2023), where the distance $d^{phys}(\cdot, \cdot)$ between nodes is their Manhattan distance (Black, 2006). We define the locality for an MC $S$ as **Locality**$(S) = L_{sem.} + L_{phys.}$, where

$$L_{sem/phys} = \frac{\Sigma_{v_1, v_2 \in S}||d^{sem/phys}(v_1, v_2)||}{\Sigma_{v \in S, u \in \mathcal{V} \setminus S}||d^{sem/phys}(v, u)||}. \tag{3}$$

The *third* criterion is **Reusability.** An MC is applicable across different tasks, enabling transfer of functionality. The reusability of an MC $S$ on a set of tasks $\mathcal{T}$ can be measured by the proportion of the tasks that $S$ appears in the LLM's circuits for those tasks, i.e.,

$$\textbf{Reusability}_{\mathcal{T}} = \frac{1}{|\mathcal{T}|}\Sigma_{T \in \mathcal{T}}\mathbb{I}(S \subseteq C_T). \tag{4}$$

The *fourth* criterion is **Composability**. When two MCs are connected, their combined operation should reflect a logical composition of their individual functions. For two MCs $S_1$ and $S_2$ with computation flow $S_1 \rightarrow S_2$, ideally we would verify that $S_1(d) = S_2^{-1}(p)$ for any input $d$ and the output $p$ given by ground-truth logical computation. This would ensure that $S_1$ provides exactly the information that

$S_2$ needs to produce the desired output. However, since both the ground-truth output $p$ and the inverse function $S^{-1}$ are not accessible, we instead evaluate composability using the expected mutual information between $S_1(d)$ and $S_2(S_1(d))$:

$$\textbf{Composability}_{\mathcal{T}}(S_1 \rightarrow S_2) = MI(S_1(d), S_2(S_1(d))). \tag{5}$$

Here, the data input $d$ complies with the distribution $\mathcal{D}$.

The *fifth* criterion is **Globality.** The total number of unique MCs and the number used in any individual circuit must stay below certain thresholds $N$ and $M$, respectively, ensuring a manageable vocabulary size. With those criteria, we formally define the MC vocabulary Discovery problem:

**Definition 3.1.** (MC vocabulary Discovery) Given (1) an LLM, (2) a set of tasks $\mathcal{T}$ within a task domain (e.g., tasks in *medicine* such as clinical diagnosis prediction), (3) a dataset $\mathcal{D}$ for each task, find a MC vocabulary that optimizes/satisfies all evaluation criteria.

# 4. ModCirc: The Proposed Methodology

In the previous section, we have formulated the novel problem of MC vocabulary discovery. Here, we present our proposed ModCirc framework which effectives solves the MC vocabulary discovery problem.

## 4.1. Model Overview and Design Motivation

In general, ModCirc finds a vocabulary of MCs with three steps, as illustrated in Fig. 2. In the first step (Section 4.2), we preprocess and unify all tasks into classification tasks, enabling us to identify SCS in the LLM using causal tracing techniques. In the second step (Section 4.3), we utilize a reinforcement learning (RL) based graph partitioning strategy to discover the modular SCSs from the SCSs of the given tasks through optimizing the MC vocabulary evaluation criteria. Finally (Section 4.4), we design an LLM-based FI prompting strategy to explain the function of each discovered modular SCS. Till then, we have acquired our MC vocabulary where each MC is a modular SCS with its FI.

## 4.2. Task Format Unification and SCSs Collection

As in Definition 3.1, we are provided with an LLM and a set of tasks $\mathcal{T}$ and their corresponding datasets $\{\mathcal{D}_T\}_{T \in \mathcal{T}}$. However, some tasks may not be suitable for circuit discovery because circuit discovery requires the desired output for each input text to be *only one token* in length. This ensures that the logit difference *on the first decoded token* caused by patching the computational node (Wang et al., 2023; Goldowsky-Dill et al., 2023) faithfully reveals the causal effect of the node. To solve the issue, we uniformly convert the formats of different tasks (e.g., medical diagnosis and medical documents summarization in the medical domain)

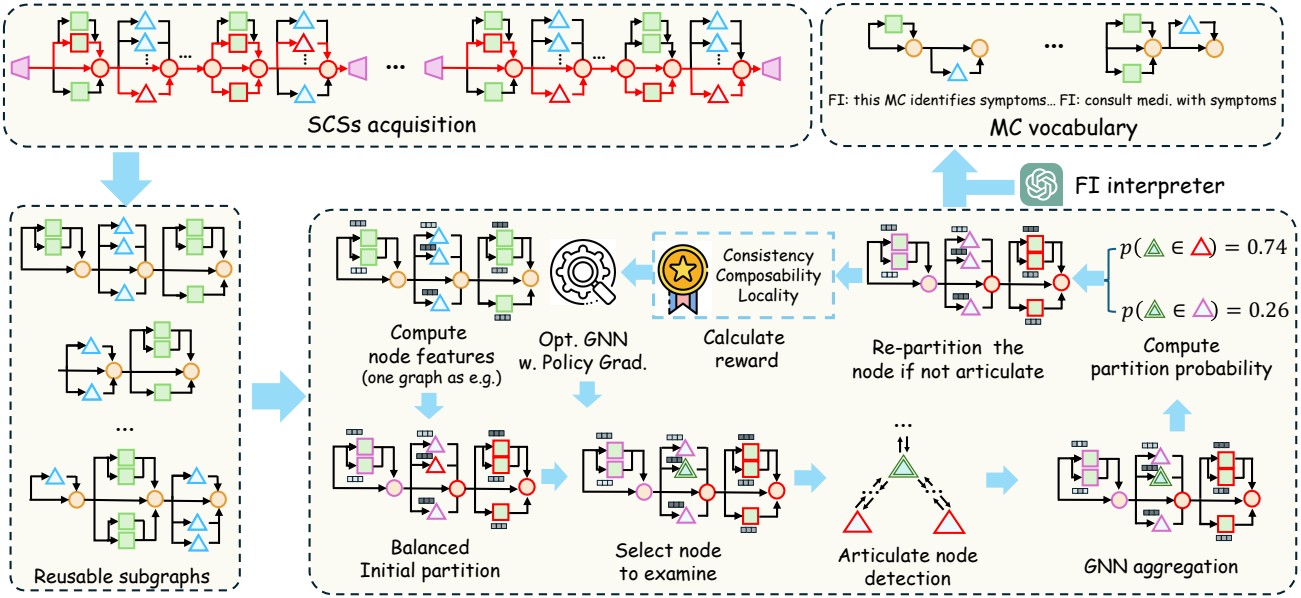

*Figure 2.* Overview of our proposed MC vocabulary discovery framework ModCirc.

into multi-choice question answering. Specifically, for a data sample consisting of an input text and a desired output text, we compose two other output choices by prompting a capable LLM (e.g., GPT-4) for similar but incorrect output.

After task format unification, we proceed to find the SCSs for all tasks on the examined LLM. Since we need to compute the SCSs for all the tasks on the LLM, performing activation patching can be very time consuming. Therefore, we adopt edge attribute patching (EAP) (Syed et al., 2023; Marks et al., 2024). EAP is the linear approximation (i.e., first-order Taylor expansion) of the activation patching formula at the output of the interested computational node with the clean input. Specifically,

$$\begin{aligned}&\text{LLM}(d_i^{clean}|do(v = v_i^{patch})) - \text{LLM}(d_i^{clean}),\\&= \nabla_v\text{LLM}|_{v=v^{clean}}(v^{patch} - v^{clean}).\end{aligned} \quad (6)$$

This approximation significantly reduces the computation of finding circuits since, originally, we need to perform two LLM inferences for each node. Now, we only need two LLM inferences and one backward propagation to compute the indirect causal effect for all of the nodes.

However, the estimation of the indirect effect given by EAP can degrade when $v^{patch}$ is not close enough to $v^{clean}$ (by definition of first-order Taylor expansion). To counter the estimation degradation, we utilize integrated gradient (IG) (Sundararajan et al., 2017; Hanna et al., 2024b; Marks et al., 2024), which instead estimates the indirect effect with first-order Taylor expansion at all $m$ equally spaced points

(i.e., $\{v_i^{mid} = v^{clean} \cdot \frac{i}{m} + v^{patch} \cdot \frac{m-i}{m}\}_{i=1}^m$), i.e.,

$$\begin{aligned}&\text{LLM}(d^{clean}|do(v = v_i^{patch})) - \text{LLM}(d^{clean}),\\&= \Sigma_i[\nabla_v\text{LLM}|_{v=v_i^{mid}}(v_i^{mid} - v_{i+1}^{mid})].\\&= [\Sigma_i\nabla_v\text{LLM}|_{v=v_i^{mid}}](v^{patch} - v^{clean}).\end{aligned} \quad (7)$$

Finally, we collect all nodes in the LLM that have the top ten values for their indirect effect in each transformer layer and connect them with all possible edges in the LLM computation graph as the discovered SCS for the examined task.

### 4.3. Modular SCSs Discovery with Neural Partitioning

With the SCSs of all tasks achieved, we proceed to discover the modular SCSs from them, as illustrated in the bottom right of Fig. 2. Advised by the reusability property of MCs (see Section 2), we first conduct reusable subgraphs generation to find the subgraphs that appear in at least two SCSs by conducting pair-wise intersection for each pair of the SCSs. This procedure guarantees that the modular SCSs are reused across the tasks while ensuring the integrity of potential MCs (see details in Section 4.3.1). Then, we design an RL-based neural partitioning method to partition each of the reusable subgraphs to become modular SCSs ccording to our proposed evaluation criteria.

Due to the complexity of the RL-based neural partitioning method, we introduce it separately with three parts: (1) ***Node Feature Acquisition***: we craft for each node a feature vector as a preparation for the partitioning algorithm. (2) ***RL-based Neural Partitioning***: the general framework description for our designed RL-based neural partitioning method. (3) ***Objective Design***: a detailed elaboration on

how we design the RL rewards to optimize the locality, composability, and globality of the discovered modular SCSs.

### 4.3.1. REUSABLE SUBGRAPH GENERATION

In Section 2, MCs must maintain reusability, meaning that each modular SCS should be able to appear in SCSs of multiple tasks. While frequent subgraph mining methods, like gSpan (Yan & Han, 2002), could identify subgraphs that appear across multiple circuits, these methods present a significant limitation: they identify frequent subgraphs of varying sizes, with smaller subgraphs typically appearing more frequently (the smallest being just two nodes connected by one edge). These small frequent subgraphs *risk fragmenting the modualr SCSs*, as they may represent only partial components rather than complete modular SCSs.

To preserve the integrity of modular SCSs, we instead unify all the circuits and preserve any overlapping components that have more than one node. This approach finds the reusable subgraph between circuits , ensuring that potential modular SCSs remain intact. However, this procedure leads to a situation that a reusable subgraph may contain multiple modular SCSs. Therefore, we follow with a partitioning procedure to differentiate individual modular SCs.

### 4.3.2. NODE FEATURE ACQUISITION

We construct node features for reusable subgraphs by combining semantic and physical features. For semantic features, we collect each node's activation patterns across all data samples and tasks. Since different node types (like attention heads) have different dimensions, we use separate MLPs to project their activations into a common dimensional space. The semantic feature for each node is the mean vector of these processed activations across all samples and tasks. For physical features, we use Euclidean grid embedding, mapping each node to coordinates based on its transformer layer and position (attention head/feed forward neuron index). The final node feature concatenates both embeddings, capturing both functional behavior and structural position to optimize modular SCS localities.

### 4.3.3. RL-BASED NEURAL PARTITIONING

In the Section 4.3.2, we equip each reusable subgraphs with node features. Given the reusable subgraphs and their node features, our motivation is to train a graph neural network (GNN) to partition the reusable subgraphs, whose output logits for each node represent the probability of the node belonging to various partitions. Furthermore, the GNN should be trained with a loss function that optimizes our proposed evaluation criteria. However, since the optimization space (i.e., all possible partitionings) is discrete, the loss functions are non-differentiable. This makes it impossible to train the GNN in traditional supervised manner. Fortunately, we may

---

**Algorithm 1** Initial Partitioning of Reusable Subgraphs

**Input** : $\mathcal{G} = \{G_1, G_2, ..., G_n\}$: Set of reusable subgraphs; $G$: the reusable subgraph to be partitioned;
**Output** : $P$: Initial partitioning for each $G \in \mathcal{G}$;
$G_{int} \leftarrow \texttt{FindIntersectionGraph}(\mathcal{G})$;
$C \leftarrow \texttt{WeaklyConnectedComponents}(G_{int})$;
$P \leftarrow \texttt{AssignPartation}(G, C)$;
$U \leftarrow V(G) \setminus C$     // Unassigned nodes in $G$
**while** $U \neq \emptyset$ **do**
  **foreach** *partition* $p \in P$ **do**
    $v \leftarrow \texttt{Random}(p)$   // Random node from partition
    $N_v \leftarrow$ neighbors of $v$ in $U$
    **if** $N_v \neq \emptyset$ **then**
      $u \leftarrow \texttt{Random}(N_v); U \leftarrow U \setminus \{u\}$   // Assign $u$ to partition of $v$
    **else if** $\texttt{Size}(p) <$ *max partition size in $N_v$'s partitions* **then**
      $u \leftarrow$ neighbor with largest partition;   // Assign $u$ to $v$'s partition
    **end**
  **end**
**end**

---

circumvent this non-differentiability issue by policy gradients (Kakade, 2001; Shah et al., 2024), an RL technique, which allows us to optimize the GNN as a policy network by sampling node partitioning actions and their corresponding rewards. Next, we specifically explain how our RL-based neural partitioning applies to a reusable subgraph.

For any reusable subgraph, we first generate an initial partitioning, as illustrated in Algorithm 1, that ensures the following properties: (1) *__internal connectivity__* within each partition (supporting physical locality in our proposed evaluation criteria); (2) *__balanced partition sizes__* across all partitions due to the initial partitioning serving as the starting point for our RL-based neural partitioning approach; and (3) *__partitioning compatible__* across all reusable subgraphs, meaning that any nodes in one partition of a reusable subgraph, if exist in another reusable subgraph, should not be separated into different partitions in that other subgraph.

To do this, we first guarantee the partitioning compatibility requirement by finding the intersection graph of all the reusable subgraphs and assigning each of its weakly connected components as a partition. Then, we expand those partitions iteratively in the reusable subgraph. Specifically, in each iteration, we randomly select a node from each current partitions and visit one of its neighbors, adding the neighbor to the same partition of the source node (for internal connectivity). If the neighbors of the source node are all assigned and the partition of the source node has a smaller size than the largest partition of its neighbor nodes, we reassign the neighbor that has the largest partition to the partition of the source node, which allows for balanced partitioning. We repeat the iteration until all nodes in the

reusable subgraph are assigned a partition.

After the initial partitioning, inspired by Shah et al. (2024), we train the GNN partitioner for each reusable subgraph upon its initial partitioning with policy gradient. As policy gradient methods operate within a Markov Decision Process (MDP) (Puterman, 1990), we will now formulate the specific MDP that our policy gradient operates within. In our MDP, decisions unfold through a sequence of states ("$s$"), actions ("$a$") and rewards ("$r$") known as *repartition trajectory* ("$\tau$"), we aim to optimize the GNN as a policy network to maximize the expected reward of repartition trajectories.

In our formulation, a repartition trajectory is a sequence of repartitioning processes, where each repartition process refers to one following procedure applied on the reusable subgraph: (1) From the currently partitioned reusable subgraph (which corresponds to state "$s$"), sample a candidate repartition node within current partition boundaries, where each node's selection probability is proportional to the fraction of heterogeneous partitions in its neighborhood; (2) Detect if the node is articulation node (Tian et al., 2017) (i.e., the node by removing which separate the nodes within a partition to isolated connected components); (3) If the node is not an articulation node, repartition the selected node by current GNN partitioner (which corresponds to action "$a$"). The reward "$r$" (Shah et al., 2024) of the repartitioning is

$$r(a) = \frac{Obj(G, P^{before}) - Obj(G, P^{after})}{Obj(G, P^{before}) + Obj(G, P^{after})}, \quad (8)$$

where $Obj(G, P^{before})$ and $Obj(G, P^{after})$ are the objective functions designed with our proposed MC vocabulary evaluation criteria, which will be specified in Section 4.3.4.

With each repartitioning process's reward specified, the comprehensive reward of the repartition trajectory $\tau$ is hence the discounted summation (by the discount factor $0 < \gamma < 1$) across the horizon of length $L > 0$

$$R(\tau) = r(a_0) + \gamma r(a_1) + \cdots + \gamma^{L-1} r(a_{L-1}). \quad (9)$$

Therefore, denote our GNN partitioner as $\pi(\theta)$ where $\theta$ is its parameters, our optimization target can be written as

$$J(\theta) = \mathbb{E}_\pi(R(\tau)). \quad (10)$$

Here, we adopt Markov Chain Monte Carlo (MCMC) (Hastings, 1970) to estimate the REINFORCE gradient (Williams, 1992) of Equation 10 for optimization of the GNN.

### 4.3.4. OBJECTIVE DESIGN

According to Section 4.3.3, we view the GNN partitioner as a policy network and optimize it with policy gradient. In the process, one significant part that informs the optimization direction is the objective in the reward function (i.e.,

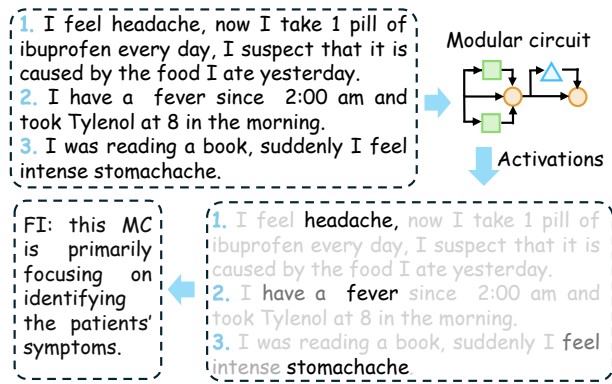

*Figure 3.* FI generation for the MCs.

Equation 8). Here, we design our modular SCS partitioning objective with the evaluation criteria specified in Section 3.

Here, we incorporate the evaluation criteria into the objective function by: (1) ***Consistency***: as quantifying semantic consistency requires the LLM prompting efforts which is rather financially and time intensive, we do not incorporate this metric into the training stage; (2) ***Locality***: we calculate the physical and semantic locality exactly by the Equ. 3, where the physical locality computation only considers the node's layer index in the model. (3) ***Reusability***: this metric was previously addressed in Section 4.3.1 during the generation of reusable computational subgraphs. We exclude it from the current objective function as a compromise in our optimization approach, accepting a lower reusability threshold for modular SCSs. (4) ***Composability***: we compute the Equ. 5 with KDE (Rosenblatt, 1956) estimation for each pair of adjacent modular SCSs and take their average value as an evaluation of the composability of the partitioning. (5) ***Globality***: as described in Section 4.3.3, we set a maximum threshold for partition numbers and a minimum size requirement relative to their containing circuits. These constraints, combined with our selection of only the top $N$ modular SCSs based on their metric values, ensure globality.

### 4.4. Functional Interpretation (FI) Generation

To complete MCs discovery, we perform FI generation as illustrated in Fig. 3 for each modular SCSs by utilizing the state-of-the-art LLM such as GPT-4 (Achiam et al., 2023) (referred in the following as "FI generator"), along with an integrated dataset which is the combination of all training data. For each data item in the integrated dataset, we process it as input to the examined LLM and extract its activation for each modular SCS. We calculate the token-wise $l_2$ norm of the activation, which creates a token-level heatmap for the data item. Then, we feed these heatmaps to the FI generator to determine the function of certain modular SCS. Notably, we may expand the text dataset for FI generation to any text datasets within the examined task domain.

# 5. Experimental Evaluations

In this section, we discover the MC vocabulary of Med-LLaMA (8B) (Labs, 2024), an LLM finetuned on LLaMA (8B) with medical domain context, when it performs medical tasks by our ModCirc. Specifically, we investigate **RQ1:** How well can ModCirc find the MCs compared with other baselines? **RQ2:** To what extent each component of ModCirc contributes to the overall performance? How will the choice of the hyperparameters affect the performance of ModCirc? **RQ3:** How does an MC involve different task circuits with consistent functioning? **RQ4:** How well can FIs of MCs transfer to new tasks? Please refer to the Appendix for supplementary results of parameter analysis and MC examples.

## 5.1. Experimental Settings

**Datasets** For training, we use "MEDAL" (Wen et al., 2020) for multi-class medical abstract classification, "Medical Abstracts" (Schopf et al., 2023) for patient condition classification from abstracts, "MedMCQA" (Pal et al., 2022) for medical multiple-choice question answering, and "Symptom to Diagnosis" (Gretel.ai, 2022) for mapping symptom to diagnoses. For evaluation, we employ four datasets with distinct tasks: "MedStatus" identifies medication status in clinical text, "MedAttr" recognizes medical entities, "Coreference"(Agrawal et al., 2022; Moon et al., 2014) resolves clinical coreference relations, and "PubMed Summ."(Cohan et al., 2018) evaluates biomedical abstract summarization.

**Baselines** Our baselines include (1) *Random*. We randomly select nodes from the circuits of each dataset and find their ego graphs with a radius of three. (2) *Frequent Random*. We propose a variant of the Random baseline by first finding the reusable subgraphs as in ModCirc. Then, we perform Random on the reusable subgraphs to obtain frequent random MCs. (3)*Kmeans*. Similarly, we first find the reusable subgraphs and then perform KMeans clustering to partition the subgraphs into MCs. (4) *Activation*. We assume that if two computational nodes are in an MC, then they would have high activation values on the same inputs. Thus, we compute the top 10 activation values for each computational node across all data samples. Then, we group nodes together if their activated samples intersect and find the connected components within the node groups.

## 5.2. Quantitative Analysis

In this section, we aim to answer **RQ1** and **RQ2**. In Table 1, we measure the composability, consistency, and reusability of the MCs identified by ModCirc and the baselines on four evaluation datasets. We observe that ModCirc, in general, achieves the best performance, especially in the reusability criterion, proving the effectiveness of our framework. Mod-

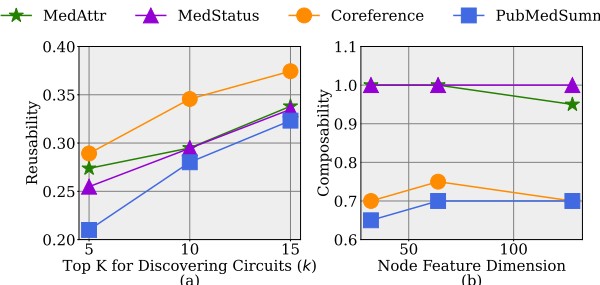

*Figure 4.* Parameter analysis.

Circ demonstrates notably superior reusability scores (0.32-0.37) across all datasets compared to baselines whose scores are below 0.21. While ModCirc maintains competitive composability scores between 0.70-and 1.00, it does not consistently outperform other approaches like Clust. and Random, which sometimes achieve perfect scores. In terms of consistency, ModCirc performs comparably to other methods, with scores ranging from 0.26-0.40, indicating reliable but not necessarily dominant performance in this aspect. The slightly lower consistency and composability scores can be attributed to ModCirc's requirement to maintain connected partitions.

We explore the effectiveness of different components in ModCirc through three variants. "ModCirc-NCirc" randomly selects circuits from the LLM's computational graphs. "ModCirc-NInit" skips the initial partitioning step and directly applies reinforcement learning-based reusable subgraph partitioning. "ModCirc-NNeuro" only performs initial partitioning, omitting the reinforcement learning-based reusable subgraph partitioning step. In Table 2, we show the performance of our ModCirc and the three variants on the MediAttr dataset. We can observe that our ModCirc possesses a consistent performance gain compared to all three variants w.r.t. consistency. Composability decreases, probably due to the additional condition of finding connected MCs. We have similar observations in other datasets.

Additionally, we examine the parameter sensitivity of our ModCirc by varying two most significant hyperparameters. Specifically, we adopt "topk ($k$) nodes", which is the number of selected nodes in each transformer layer for SCS construction (see Section 4.2); and "node feature dimension" referring to the node feature dimensions (see Section 4.3.2). In Fig. 4 (a) shows that increasing the $k$ leads to improved reusability (from 0.20 to 0.40) across all datasets, with MedAttr and PubMedSumm most notable. Fig. 4 (b) shows that composability first increases then decreases with the increase of node feature dimension (peak at 64). Other criteria are stable by varying these hyperparameters.

*Table 1.* Effectiveness of our ModCirc and baselines in finding MCs. The optimal results are in **bold**, and the runner-up are underlined.

| Datasets | Metrics | Act. | Clust. | Freq. | Random | ModCirc |
|---|---|---|---|---|---|---|
| Coreference | Composability | 0.75 ± 0.00 | **0.80 ± 0.10** | 0.65 ± 0.12 | 0.75 ± 0.00 | 0.70 ± 0.10 |
|  | Consistency | 0.35 ± 0.03 | 0.39 ± 0.02 | 0.39 ± 0.02 | 0.37 ± 0.02 | **0.40 ± 0.02** |
|  | Reusability | 0.06 ± 0.00 | 0.13 ± 0.01 | 0.21 ± 0.01 | 0.11 ± 0.01 | **0.37 ± 0.01** |
| MediAttr. | Composability | **1.00 ± 0.00** | **1.00 ± 0.00** | **1.00 ± 0.00** | **1.00 ± 0.00** | 0.95 ± 0.10 |
|  | Consistency | 0.32 ± 0.01 | 0.34 ± 0.01 | **0.35 ± 0.01** | 0.33 ± 0.02 | 0.33 ± 0.01 |
|  | Reusability | 0.06 ± 0.01 | 0.11 ± 0.01 | 0.18 ± 0.01 | 0.10 ± 0.01 | **0.34 ± 0.01** |
| MediStatus | Composability | **1.00 ± 0.00** | **1.00 ± 0.00** | **1.00 ± 0.00** | **1.00 ± 0.00** | **1.00 ± 0.00** |
|  | Consistency | 0.24 ± 0.01 | **0.26 ± 0.00** | **0.26 ± 0.01** | **0.26 ± 0.01** | **0.26 ± 0.01** |
|  | Reusability | 0.05 ± 0.01 | 0.10 ± 0.01 | 0.17 ± 0.01 | 0.09 ± 0.01 | **0.33 ± 0.00** |
| PubSumm. | Composability | **0.75 ± 0.00** | **0.75 ± 0.00** | **0.75 ± 0.00** | 0.70 ± 0.10 | 0.70 ± 0.10 |
|  | Consistency | 0.31 ± 0.02 | 0.33 ± 0.02 | 0.34 ± 0.01 | 0.33 ± 0.02 | **0.34 ± 0.03** |
|  | Reusability | 0.05 ± 0.00 | 0.10 ± 0.01 | 0.18 ± 0.01 | 0.10 ± 0.00 | **0.32 ± 0.01** |

*Table 2.* Performance comparison of ModCirc variants specified in Section 5.2 on MediAttr. "Compo.", "Const." and "Reusa." represent composability, consistency, and reusability, respectively.

| Method | Compo. | Const. | Reusa. |
|---|---|---|---|
| ModCirc | 0.95 ± 0.10 | 0.33 ± 0.01 | 0.34 ± 0.01 |
| ModCirc-NCirc | 1.00 ± 0.00 | 0.30 ± 0.01 | 0.20 ± 0.00 |
| ModCirc-NInit | 1.00 ± 0.00 | 0.32 ± 0.01 | 0.34 ± 0.01 |
| ModCirc-NNeuro | 0.95 ± 0.10 | 0.31 ± 0.01 | 0.34 ± 0.01 |

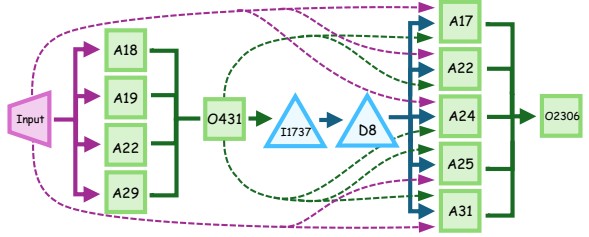

*Figure 5.* An example MC spanning two layers. Green squares show self-attention components (A: attention heads, O: self-projection neurons) while blue triangles are feed-forward networks (I: MLP input neurons, D: MLP down-projection neurons).

## 5.3. Qualitative Analysis

In this section, we demonstrate the quality of our MC vocabulary with a case study to answer **RQ3**. In Fig. 5, we illustrate a representative MC, which provides compelling evidence for the effectiveness and semantic consistency of the identified MC. This MC is highly reusable, since it appears in SCSs of three out of four evaluation tasks (MedAttr, MedStatus, and Coreference), strongly suggesting its fundamental role in medical language understanding.

The MC's FI, which focuses on "identifying and classifying biological entities and interactions, particularly in health and disease," demonstrates remarkable alignment with the semantic requirements of these three tasks. For medication attribute identification, the MC's ability to recognize biological entities serves the task of identifying medications and their characteristics. For medication status classification, the MC's capability to analyze "complex biological

data" supports determining medication statuses, while in coreference resolution, its specialization enables accurate tracking of medical terms across text. The absence of this MC in PubMed Summ is reasonable, as this task focuses more on high-level summarization rather than specific entity identification. This semantic alignment between the MC's FI and its usage across tasks validates interpretation.

## 5.4. MC Transferability and Composability

In this section, we address **RQ4**. We test ModCirc's transferability and composability by training ModCirc on GPT-2 Small (Radford et al., 2019) with four training NLP tasks: AGNews (Zhang et al., 2015), MPQA (Wiebe et al., 2005), Universal Dependency (Silveira et al., 2014), and TREC (Voorhees & Tice, 2000). We then generate MCs for two new unseen tasks whose ground-truth circuits (including FIs) are established in existing work: indirect object identification (IOI) (Wang et al., 2022) and acronym detection (García-Carrasco et al., 2024).

The IOI task aims to predict the indirect object (IO) in a sentence where the subject (S) appears twice. For example, in the sentence "Mary (IO) and Amy (S) met, and Amy (S) handed a pen to __," the blank represents the position where an indirect object should be identified based on the sentence context. The acronym detection task (García-Carrasco et al., 2024) predicts the acronym of a three-word phrase (e.g., the acronym of "Limited Liability Company" is LLC).

For the IOI task, ModCirc recovers 92% (23/25) of the ground-truth circuits' components, and Fig. 6 shows the analyzed MCs. We observe from Fig. 6 that the FIs of the MCs generated by ModCirc correspond to the FIs of the ground-truth circuit (Wang et al., 2022), indicating the transferability of ModCirc. Specifically, MC 0 identifies two Induction Heads, whose FI is to highlight prior occurring subjects (Wang et al., 2022), complementing the MC's FI of "identifying specific entities," such as nouns. MC 1 contains two Name Mover (NM) Heads that "attend to previous names" and one S-Inhibition (SI) Head that "removes duplicate names identified by the NM Heads" (Wang et al.,

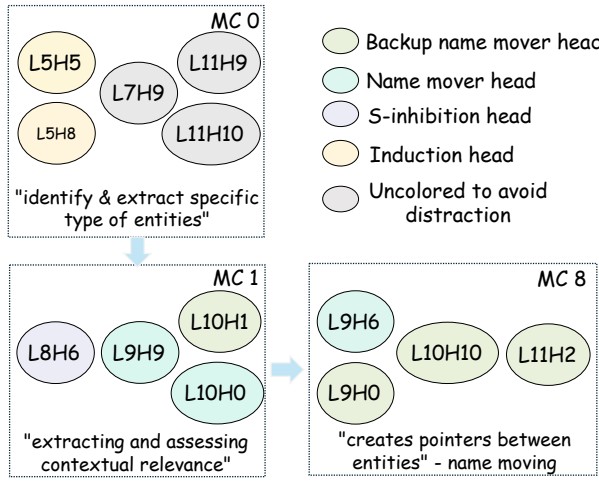

*Figure 6.* MCs identified by ModCirc for IOI. "L" and "H" represent the layer and head indices, respectively.

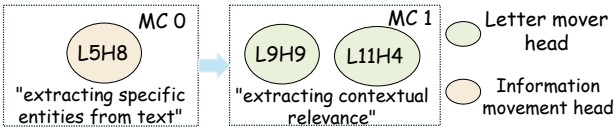

*Figure 7.* MCs identified by ModCirc for acronym detection.

2022) [2]. This aligns with the MC's FI of "extracting and assessing contextual relevance," where NM Heads extract names, and SI Heads assess and remove duplicates. MC 8 contains one NM Head and three Backup Name Mover (BNM) Heads. Unlike MC 1, MC 8's FI is to "create associations or pointers between entities and their attributes," which matches the NM Head's role to associate and output the indirect object. Furthermore, the FIs of the MCs exhibit composability: MC 0 detects duplicate tokens, MC 1 assesses and removes duplication, and MC 8 outputs answers, matching the existing ground-truth circuit structure.

For the acronym detection task, ModCirc recovers 87.5% ground-truth circuit nodes (7/8), and the analyzed MCs are shown in Fig. 7. We observe from Fig. 7 that ModCirc produces FIs that align with the ground-truth FIs (García-Carrasco et al., 2024): (1) MC 0 captures Head L5H8, and its FI "extracting entities" agrees with the head's FI "find capital letter"; (2) MC 1 contains letter mover (LM) heads L9H9 and L11H4, aligning the MC's FI "assessing context" with head's ground-truth FI "copy capital". The "assess" here implies the copying procedure. Additionally, MC 0 identifies the capital letter position, and MC 1 copies the capital letter to answer position, composing the ground-truth circuit's FI (García-Carrasco et al., 2024).

---

[2]Name Mover (NM) Heads, S-Inhibition (SI) Heads and the Backup Name Mover (BNM) Heads (referred in the following texts) are defined and identified by existing work.

# 6. Related Work

## 6.1. Mechanistic Interpretability

Mechanistic interpretability (MI) reverse-engineers components of machine learning models into comprehensible concepts, allowing for a deeper understanding of model behavior (Bereska & Gavves, 2024; Rai et al., 2024; Singh et al., 2024). Recent works in MI concentrate on identifying end-to-end circuits, pathways that trace the flow of information from the inputs to the outputs (Conmy et al., 2023; Wang et al., 2022; Hanna et al., 2024a). However, these circuits are task-specific, raising questions about their applicability to other tasks. Some studies attempt to address this limitation by investigating the generalizability of existing circuits. For instance, researchers have tested the performance of the indirect object identification (IOI) circuit on variations of the original task (Nainani et al., 2024) and examined its reusability in a different context (Merullo et al., 2023). Building on these efforts, this work aims to compile an MC vocabulary that can be reused across various tasks.

## 6.2. LLM Modularity

LLM modularity refers to the integration of multiple components, each specializing in a specific task, to construct an LLM with diverse functionalities (Munikoti et al., 2024; Shao et al., 2024; Kaddour et al., 2023). One approach emphasizes a model-centric perspective, designing components to specialize in specific tasks (Xiao et al., 2024; Feng et al., 2024; Ostapenko et al., 2024; Muqeeth et al., 2024; Kang et al., 2024). Another approach focuses on the training process, encouraging the development of modularity during training (Liu et al., 2023; Nainani, 2024). However, these methods incur high training costs, posing a significant challenge for LLMs. Our work addresses this issue by examining pre-trained models for modularity and applying causal mediation analysis to uncover modular circuits.

# 7. Conclusion

We take an initial step towards global-level mechanistic interpretability (MI) which address the task-specific and high computational costs limitations of existing MI methods. First, we formulate a novel problem of discovering a modular circuit (MC) vocabulary. Then, we propose ModCirc, a principled method for MC vocabulary discovery that addresses both limitations. Finally, we examine ModCirc on an LLM and successfully discovered its MC vocabulary with extensive quantitative and qualitative evidence.

# Impact Statement

This work on modular circuit discovery represents a positive direction for AI research, as it aims to enhance our

understanding of how language models process information through purely observational analysis of existing systems. By improving model interpretability and efficiency through identifying reusable components, this research could lead to more reliable and resource-efficient AI systems while reducing computational costs. The approach enables better auditing and validation of AI behavior through enhanced transparency, which aligns with broader goals of developing trustworthy AI technology that can benefit society. Since the research focuses on analyzing already-trained models rather than developing new capabilities, it contributes constructively to scientific knowledge while maintaining a responsible approach to AI advancement.

## Acknowledgement

This work is supported in part by the National Science Foundation (NSF) under grants IIS-2006844, IIS-2144209, IIS-2223769, CNS-2154962, BCS-2228534, and CMMI-2411248; the Office of Naval Research (ONR) under grant N000142412636; and the Commonwealth Cyber Initiative (CCI) under grant VV-1Q24-011.

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

# A. Limitation

The proposed ModCirc framework, while demonstrating promising results in discovering modular circuits within large language models, has several important limitations that should be acknowledged: First, our current implementation relies heavily on the quality and diversity of training task circuits. The discovered modular circuits (MCs) are inherently limited by the scope and variety of the training tasks used. If the training tasks do not adequately cover certain functional aspects of the model, the resulting MC vocabulary may miss important modular components that could be present in the model but not activated by our chosen tasks. Second, the evaluation of consistency and reusability across tasks remains challenging. The current metrics may not fully capture the nuanced ways in which circuits might be reused or adapted across different contexts, potentially leading to an oversimplified understanding of the model's modularity.

# B. Implementation Details

ModCirc is implemented in PyTorch (Ansel & Yang, 2024) using the Transformers package (Wolf et al., 2020) to access MedLlama and the PyTorch Geometric package (Fey & Lenssen, 2019) for constructing the GNN model. We also used the OpenAI API to access GPT-4o-Mini (OpenAI, 2025) for the functional interpretation and evaluation phase. In our experiments, the learning rate is 0.001, and the training epoch number is 10. Experiments are carried out on an Nvidia RTX A6000, and the reported numerical results are averaged across five different runs.

# C. Supplementary Experiment Results

### C.1. Parameter Analysis

In this subsection, we present additional parameter analysis of our proposed ModCirc. We observe from Fig. 8 that there exist four steady relationships: composability and reusability are steady with the increase of top $k$ discovering circuits and node feature dimensions, respectively. Besides, consistency remain intact with the increase of both the top $k$ discovering circuits and node feature dimensions.

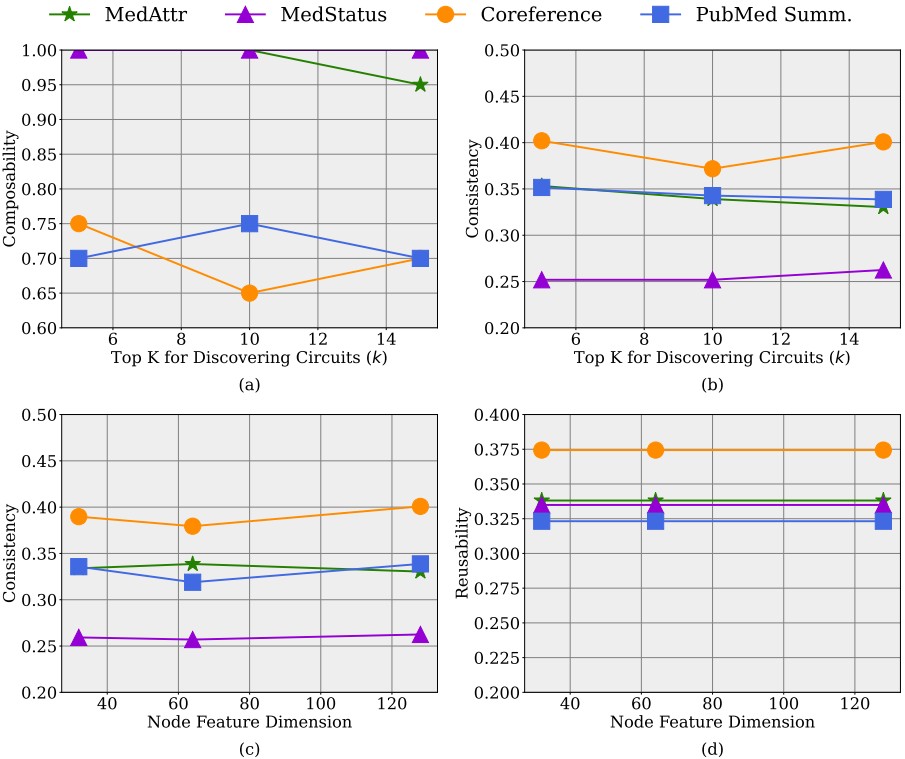

*Figure 8.* Additional parameter analysis of our proposed ModCirc.

## C.2. Example Modular Circuits

We show two example modular circuits as in Fig. 9. For complete set of the modular circuits, see https://github.com/YinhanHe123/ModCirc.

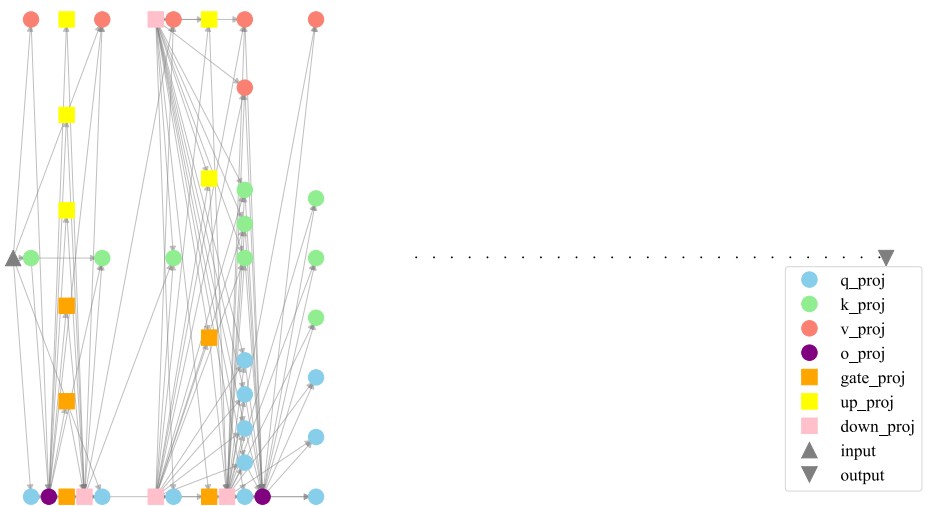

Modular Circuit 1. This MC specializes in infectious disease classification from symptom patterns

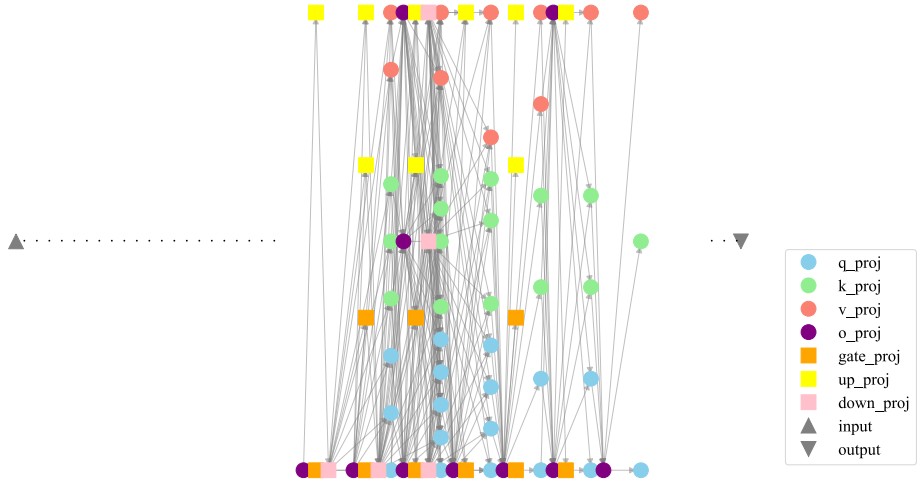

Modular Circuit 2. This MC focuses on respiratory and neurological condition diagnosis

*Figure 9.* Modular Circuit 1. This MC specializes in infectious disease classification from symptom patterns

### C.3. Functional Interpretations of the Modular Circuits

We show two example modular circuits' functional interpretations as in Table 3. For complete set of the functional interpretations, see https://github.com/YinhanHe123/ModCirc.

*Table 3.* Descriptions of Modular Circuits and Their Functions

| Circuit | Description |
|---|---|
| 0 | The main function of modular circuit 0 appears to be the classification and differentiation of medical conditions based on presented symptoms and clinical information. High importance scores are associated with texts that describe specific symptoms (e.g., runny nose, fever, itching) and medical conditions (e.g., common cold, malaria, renovascular hypertension). The circuit likely focuses on identifying patterns in symptomatology and diagnostic tests to accurately categorize various health issues, particularly in the context of infectious diseases and conditions related to organ function. |
| 1 | The main function of modular circuit 1 appears to be the classification and diagnosis of medical conditions based on symptom descriptions and clinical contexts. The high importance scores are associated with texts that detail specific symptoms (e.g., cough, pain, stiffness) and treatments (e.g., chemotherapy regimens), indicating that the circuit is designed to interpret and categorize health-related information effectively. It focuses on identifying patterns in symptoms and their potential links to specific diseases or conditions, particularly in respiratory, neurological, and musculoskeletal contexts. |

### C.4. Ablation Study

Here, we demonstrate the extended ablation results on additional datasets in Table 4 . These results show that ModCirc consistently outperforms (up to 5%) its ablation variant without RL-based subgraph partitioning. Note that "Reusability" is excluded from this comparison since it is primarily influenced by the initial partitioning rather than the RL-based subgraph partitioning.

*Table 4.* Comparison of ablation and original results across datasets

| Dataset | Metric | Ablation Results | Original Results |
|---|---|---|---|
| Coreference | Composition | $0.65 \pm 0.12$ | $0.70 \pm 0.10$ |
| | Consistency | $0.37 \pm 0.01$ | $0.40 \pm 0.02$ |
| PubmedSumm | Composition | $0.65 \pm 0.12$ | $0.70 \pm 0.10$ |
| | Consistency | $0.34 \pm 0.01$ | $0.34 \pm 0.03$ |

