# OpenReview forum: "Towards Global-level Mechanistic Interpretability: A Perspective of Modular Circuits of Large Language Models"
_ICML.cc/2025/Conference — ICML 2025 poster_

### Official Review · Reviewer_4yNw · 2025-03-04

**Overall Recommendation:** 3

**Summary:**

The paper introduces ModCirc, a framework for global-level mechanistic interpretability of LLMs by discovering modular circuits -- task-agnostic functional units that enable cross-task interpretability while reducing computational costs. It defines the MC vocabulary discovery problem with five evaluation criteria: consistency, locality, reusability, composability, and globality. The proposed reinforcement learning-based graph neural partitioning method identifies reusable computational subgraphs and partitions them into modular circuits, which are then assigned functional interpretations. Experiments on MedLLaMA-8B across diverse medical tasks demonstrate that ModCirc effectively identifies modular circuits with strong reusability and competitive composability and consistency, enabling scalable and interpretable LLMs.

**Claims And Evidence:**

Claim1 -- Functional Interpretations Accuracy: The paper relies on GPT-4-generated FIs, but does not rigorously evaluate their correctness.

Claim2 -- Scalability to General LLMs: All experiments focus on MedLLaMA-8B, leaving open the question of whether ModCirc generalizes to larger or non-medical models.

Claim3 -- Composability and Consistency Improvements: ModCirc does not always outperform baselines on these metrics, suggesting that task-agnostic modular circuits might not always maintain strong interpretability across different contexts.

**Essential References Not Discussed:**

None

**Experimental Designs Or Analyses:**

The study focuses only on MedLLaMA-8B, a domain-specific model, and does not test whether ModCirc works for general-purpose LLMs.

Claims about scalability and generalization should be validated on non-medical tasks.

Functional Interpretations are not rigorously evaluated.

**Methods And Evaluation Criteria:**

Yes, the proposed methods and evaluation criteria align well with the problem of global-level mechanistic interpretability in LLMs. However, the evaluation is limited to MedLLaMA-8B and medical NLP tasks, which restricts generalizability. Moreover, functional interpretation quality is not rigorously validated beyond GPT-4 outputs,

**Other Comments Or Suggestions:**

(1) Provide runtime comparisons between ModCirc and baselines

(2) Explain why ModCirc does not always outperform baselines in composability and investigate methods to improve functional stability across tasks.

**Other Strengths And Weaknesses:**

Strengths:

(1) The paper introduces a novel formulation of modular circuit discovery, extending mechanistic interpretability research by making circuits reusable across tasks rather than task-specific.

(2) The paper is well-structured, with clear explanations of evaluation criteria, algorithmic steps, and experimental setups, making it easy to follow.

(3) Code with a detailed readme file is provided.


Weaknesses:

(1) The experiments are restricted to MedLLaMA-8B and medical NLP tasks, limiting claims of scalability to general LLMs.

(2) The GPT-4-generated FIs are not validated by human experts, raising concerns about potential biases or hallucinations in interpretations.

(3) The reinforcement learning-based neural partitioning may be computationally expensive, and alternative differentiable or self-supervised methods could be explored for efficiency.

**Questions For Authors:**

(1) Have you tested ModCirc on general-purpose LLMs or non-medical tasks?

(2) How do you ensure the accuracy of GPT-4-generated FIs?

(3) How does the runtime of ModCirc compare to baselines?

(4) Why does ModCirc sometimes underperform baselines in composability?

**Relation To Broader Scientific Literature:**

The paper builds on mechanistic interpretability (MI) research, particularly prior work on task-specific circuit discovery (e.g., activation patching and causal tracing in LLMs) but extends it by introducing modular circuits (MCs) that generalize across tasks.

**Theoretical Claims:**

The paper does not present formal mathematical proofs but introduces a set of evaluation criteria and an algorithmic framework (ModCirc) for discovering modular circuits in large language models.

---

> ### Author Rebuttal · Authors · 2025-04-01
>
> We sincerely appreciate the time and effort you've dedicated to reviewing and providing invaluable feedback. We provide a point-to-point reply below for the mentioned concerns and questions. We use the [anonymous repository](https://anonymous.4open.science/r/ModCirc-4887/README.md) (termed "the link") to store supplementary results.
>
> > **Reviewer**: Scale the experiments to general LLMs.
>
> **Answer**: Thank you for the suggestion. We further conduct our ModCirc in GPT-2 Small (see Fig. 1 and Fig. 2); we train the ModCirc on four datasets: Trec, AGNews, MPQA, and Universal Dependency. We generate ten modular circuits (270 computational nodes) from over 50k nodes. Our generated modular circuits successfully uncover the ground-truth circuits with consistent and composable functional interpretations on tasks researched in two existing works. Due to the space limit, we kindly refer you to our response to reviewer aN6A's first question for more observations.
>
> > **Reviewer**: How to verify GPT-4 generated FIs are correct?
>
> **Answer**: We may verify it with causal intervention. Here, we show some example results for modular circuit 18 in MedLLaMA through causal intervention (see Fig. 3 and 4 in the link). Our ModCirc identifies this modular circuit as performing "dosage detection." To verify this with causal intervention, we create a test dataset. In the dataset, each data item is a sentence containing at least one word related to dosage. We corrupt the dataset in two ways: (1) masking dosage tokens and (2) masking random non-dosage tokens. Results in Fig.3 and Fig. 4 show that masking dosage tokens significantly reduces circuit 18 activation, while random token masking either maintains or increases activation levels. The increased activation with random masking may occur because shorter sequences allow the circuit to focus more effectively on dosage detection. These findings confirm our interpretation that modular circuit 18 specializes in dosage detection.
>
>
> > **Reviewer**: The FIs are not examined by human experts.
>
> **Answer**: We recognize that the FIs are not thoroughly inspected by human experts. However, due to the budget limit, we do not have access to perform the expert inspection. We plan on releasing our experiment results online and invite broad crowdsourcing support to finish the human verification.
>
> > **Reviewer**: Explain why ModCirc does not always outperform baselines in composability and investigate methods to improve functional stability across tasks.
>
> **Answer**:
> - Our approach emphasizes reusability and demonstrates significantly better performance than the baselines shown in Table 1. This meets a crucial need: a modular circuit vocabulary must be able to cover the circuits for new, unseen tasks. The baselines do not satisfy this need, as their generated vocabularies only contain a limited segment of unseen task circuits, leading to bias. Their reported consistency and composability scores appear high because they are only evaluated on smaller circuit portions, which is inherently easier to maintain consistency than entire circuits.
> - This metric depends on GPT-4o mini to yield scaled assessments of functional interpretation consistency. However, there are concerns that GPT-4o mini may produce inflated scores, an issue recognized by the community as Overconfidence Bias [1], potentially diminishing their distinctiveness.
> - We design this metric due to the absence of well-acknowledged metrics for assessing functional interpretation consistency in the field. We perceive this more as a chance for future exploration rather than a limitation of our research.
> - We think having domain experts to evaluate functional interpretation consistency would be the ideal approach to measure the consistency more faithfully. However, we are unable to finish it due to the budget limit.
>
> [1] Li, Haitao, et al. "Llms-as-judges: a comprehensive survey on llm-based evaluation methods." arXiv preprint arXiv:2412.05579 (2024).
>
> > **Reviewer**: Provide run time comparison for ModCirc and the baselines.
>
> **Answer**:
> The table below illustrates the average wall-clock time to experiment with the methods. We observe that ModCirc is about 2.5x slower than most baselines, except for "Clust.", where it is 2x slower. However, it should be considered that most baselines are heuristic methods requiring much less time than a machine learning method. Thus, we deem that the time difference is a necessary trade-off for the generally improved performance of ModCirc displays when compared to the baselines.
>
> |  Method | Time (s) |
> |:-------:|:--------:|
> |   Act.  |  941.14  |
> |  Clust. |  1321.00 |
> |  Freq.  |  957.13  |
> |  Random |  961.75  |
> | ModCirc |  2357.27 |

---

### Official Review · Reviewer_h4Pi · 2025-03-10

**Overall Recommendation:** 2

**Summary:**

This paper proposes a novel formulation of the circuit discovery problem, which is a mechanistic interpretability task concerned with identifying a small subset of an LLM’s components responsible for a specific task. The authors propose a variant of this problem that involves identifying multiple subsets of the model’s components (a circuit vocabulary) that are modular and are used in multiple different tasks. The paper defines 5 criteria to evaluate a modular circuit vocabulary: consistency, locality, reusability, composability, and globality.
The paper then presents a method to tackle the proposed problem. The method consists of (1) identifying an initial set of computational subgraphs for each task considered, (2) generating an initial partitioning of the subgraphs, (3) training a GNN to partition the reusable subgraphs using an RL-based procedure that optimizes for the proposed evaluation criteria. The method is evaluated and compared to a set of baselines on 4 tasks in the medical domain. Additionally, the authors present an ablation study and an analysis of the effect of the method’s hyperparameters.

**Claims And Evidence:**

The authors claim that a modular circuit vocabulary should exhibit consistency, among other properties. However, the empirical results indicate that the proposed method achieves consistency scores that are nearly on par with simple “random” and “frequency” baselines.  A high degree of functional consistency of a component across different tasks seems a necessary condition to claim modularity, and this is not clear from the empirical results.

**Essential References Not Discussed:**

None.

**Experimental Designs Or Analyses:**

- The experimental setting assumes that the tasks under consideration belong to a specific domain (as noted in Definition 3.1, line 117). The rationale behind this domain-specific assumption is not well explained, and the choice of the medical domain for evaluation is not adequately motivated.
- The paper relies on an auto-interpretability procedure to generate functional interpretations of components. This reliance is potentially problematic given that LLM-generated interpretations can be imperfect and may not fully capture the underlying functionality.

**Methods And Evaluation Criteria:**

- The paper defines the consistency criterion in Section 3, which relies on a “synonymity” operator computed using an LLM. However, the implementation details of this operator and the practical computation of this quantity are not clearly described.
- The ablation study in Table 2 shows only a minimal performance drop when the RL-based subgraph partitioning is removed. This result suggests that the proposed optimization procedure may not be essential for achieving the desired task performance.

**Other Comments Or Suggestions:**

- The use of abbreviations, such as “MC” for modular circuit vocabulary and “ModCirc” for the method, can be confusing at first reading.
- line 222: “modualr”

**Other Strengths And Weaknesses:**

None.

**Questions For Authors:**

1. Could you provide more details on how the “synonymity” operator is implemented, and explain the practical steps involved in computing the consistency metric using an LLM?
2. The ablation results in Table 2 show only a minimal performance drop when the RL-based subgraph partitioning is removed. Could you clarify the specific contribution of the RL component, and whether a simpler method might achieve similar performance?
3. The consistency scores achieved by ModCirc are nearly identical to those of the “random” and “frequency” baselines. How do you interpret this finding, and what additional evidence supports the claim that your method attains the high degree of functional consistency required for a truly modular circuit vocabulary?

**Relation To Broader Scientific Literature:**

The contributions of the paper connect with the increasingly larger literature about LLM circuit finding and behavior localization. This is adequately discussed by the authors in the related work section.

**Theoretical Claims:**

N/A

---

> ### Author Rebuttal · Authors · 2025-04-01
>
> We sincerely thank you for providing invaluable feedback. We provide a point-to-point reply below to address the concerns and questions. We use the [anonymous repository](https://anonymous.4open.science/r/ModCirc-4887/README.md) (termed "the link") to store supplementary results.
>
> > **Reviewer**: Explain how consistency is calculated in evaluation.
>
> **Answer**: "Consistency" is calculated by following steps:
> - For each modular circuit (MC) that intersects with the task's corresponding circuit, we use GPT-4o mini to generate a task-specific functional interpretation (FI) for it.
> - We then prompt GPT-4o mini to rate (on a scale of 1-5) how consistent this task-specific interpretation is with the corresponding FI in our MC vocabulary.
> - Finally, we calculate the mean value of these consistency scores across all MCs to determine the overall consistency for the method performed on a given dataset.
>
> > **Reviewer**: The Consistency score of ModCirc does not achieve a significant lead than baselines.
>
> **Answer**:
> - Our method prioritizes reusability, where ModCirc has much higher performance than the baselines in Table 1 in the paper. This addresses a fundamental requirement: MC vocabulary should cover the circuit of new unseen tasks. All baselines fail to meet this requirement, as their generated vocabularies only capture a small portion of the task's circuit. This creates bias—their consistency and composability scores appear high because they are only tested on small circuit segments (it is easier to be consistent on partial circuits than complete ones).
> - This metric relies on GPT 4o-mini to provide scaled scores evaluating FI consistency. However, GPT 4o-mini may inflate scores, acknowledged as Overconfidence Bias [1], making the lead less distinguishable.
> - We use this metric due to the lack of theoretically principled FI consistency metrics in the community. We view this more as an opportunity for future work rather than a limitation of our research.
> - We believe that having domain experts manually score FI consistency would be optimal. Unfortunately, budget constraints prevented us from pursuing this approach.
>
> [1] Li, Haitao, et al. "Llms-as-judges: a comprehensive survey on llm-based evaluation methods." arXiv preprint arXiv:2412.05579 (2024).
>
> > **Reviewer**: Removing RL-based subgraph partitioning leads to a low-performance drop.
>
> **Answer**: In fact, the RL-based subgraph partitioning is contributing significantly to the ModCirc's performance:
> - Please refer to the extended ablation results on additional datasets in Table 1 (linked). These results show that ModCirc consistently outperforms (up to 5%) its ablation variant without RL-based subgraph partitioning. (Note: Reusability is excluded from this comparison since it is primarily influenced by the initial partitioning rather than the RL-based subgraph partitioning.)
> - As mentioned in our previous response, GPT-4o mini tends to exhibit overconfidence bias, which may understate the actual performance gap. To better illustrate the importance of RL-based subgraph partitioning, we include a qualitative case study. Specifically, we apply ModCirc to GPT-2 Small using four training datasets and evaluate two unseen datasets: Indirect Object Identification (IOI) and Acronym Detection [2, 3]. Our results show that ModCirc recovers the ground-truth circuits with consistent and composable FI (see Fig. 1-2 in the link). However, the variant without RL-based partitioning fails to retrieve the correct circuits and produces nearly identical FIs across different MCs, making it impossible to match the ground-truth circuit's FIs.
>
> [2] Wang, Kevin Ro, et al. "Interpretability in the Wild: a Circuit for Indirect Object Identification in GPT-2 Small." ICLR, 2023.
>
> [3] García-Carrasco, Jorge, Alejandro Maté, and Juan Carlos Trujillo. "How does gpt-2 predict acronyms? extracting and understanding a circuit via mechanistic interpretability." AISTATS, 2024.
>
> > **Reviewer**: Explain why the tasks are restricted to a particular domain and why using the medical domain.
>
> **Answer**:
> - This is to guarantee the functionality transferability assumption. When two tasks are in different domains, their task semantics can significantly vary, making generalizing functionality challenging.
> - We choose the medical field as it is a high-stakes field where interpretability is significant for model deployments. Furthermore, in the medical domain, some well-performed medical specialized LLMs such as MedLLaMA and many high-quality medical language datasets are available, facilitating us to explore the LLM's behavior more precisely.

---

> > ### Comment · Reviewer_h4Pi · 2025-04-05
> >
> > I'd like to thank the authors for their response. However, I will retain my original score, as it accurately reflects my current assessment of the paper.

---

> > > ### Author Response · Authors · 2025-04-06
> > >
> > > Dear Reviewer h4Pi:
> > >
> > > We sincerely appreciate that you take time to review our rebuttal. We tried our best to carefully address each of your concerns with point-by-point responses and extensive targeted supplementary experiments. If any concerns remain unaddressed, we are ready to provide further clarification.
> > >
> > > Sincerely,
> > >
> > > Authors from Submission 8221

---

### Official Review · Reviewer_Ezqf · 2025-03-12

**Overall Recommendation:** 4

**Summary:**

This work aims to address key challenges in the current state of mechanistic interpretability literature, namely: (1) the limited generalization of results from task-specific circuit analysis and (2) the high human effort required to determine the functional interpretation of each computational node. To tackle these issues, it proposed the concept of modular circuits (MCs)—subgraphs within a model’s computation graph that are frequently utilized across different tasks. In addition to defining MCs and outlining their key desiderata, it introduced a framework for discovering MCs given a LLM and a set of tasks. Applying this framework to medical tasks, it identified MCs and found that both quantitative and qualitative evaluations show promising results.

**Claims And Evidence:**

This work presents two primary claims:
1. The formalization of a modular circuit vocabulary that defines concrete desiderata, namely Consistency, Locality, Reusability, Compositionality, and Globality.
2. The introduction of ModCirc, a framework designed to identify modular circuits that satisfy these desiderata.

Both claims are supported by evidence. Specifically, the ModCirc framework demonstrates the practical implementation of the proposed vocabulary. Furthermore, its evaluation on medical tasks highlights its effectiveness.

**Essential References Not Discussed:**

None

**Experimental Designs Or Analyses:**

The paper could be improved by incorporating following suggestions:
- The paper presents an example of a modular circuit that identifies biological entities and interactions. However, providing additional examples would strengthen the argument. Ideally, the authors could enumerate all identified MCs, perhaps in the appendix.
- I would have liked to see ablation experiments on the identified MCs. For instance, if the previously mentioned MC were removed from the model, how would it impact the model’s performance across various tasks?
  - Such an analysis would also help validate the claims in Section 5.3 regarding the usefulness of the investigated MC for different medical tasks.
- Finally, while ModCirc generally outperforms other baselines, the differences in scores are not substantial (based on results in Table 1), indicating significant room for further improvement in the proposed framework.

**Methods And Evaluation Criteria:**

Overall, while I am broadly satisfied with the soundness of the methods and evaluation criteria, the paper could be improved by incorporating the following points (in addition to those mentioned in the Experimental Designs or Analyses section):
- The proposed ModCirc framework applies Edge Attribution Patching to identify causal components. However, their filtration criterion retains the top 10 nodes at each layer, meaning that for the LLaMA-8B model, the SCS would consist of 320 components.
  - While this criterion is not entirely unreasonable, I encourage the authors to explore alternative selection methods based on existing metics such as faithfulness, minimality, and completeness. For instance, they could retain the top N nodes necessary to achieve a faithfulness score of 1.
- Additionally, functional interpretation is performed using GPT-4 through prompting. While the generated text can suggest hypotheses about the functionality of SCS components, it does not verify their correctness. Existing research often employs causal intervention techniques to validate such hypotheses. I recommend incorporating these techniques into the framework to strengthen the analysis.

**Other Comments Or Suggestions:**

List of typos:
- Section 4 (line 125): effectives -> effectively.
- Section 4.3 (line198): ccording -> according.

**Other Strengths And Weaknesses:**

- The explanation of the Locality desideratum could be improved; I initially found it difficult to grasp.
- Additionally, I encourage the authors to include a brief conclusion paragraph after Section 4.3.4 and before Section 4.4. This summary would help reinforce the key points of Section 4.3, particularly for readers who may not be familiar with GNNs and/or RL.
- Section 2 (line 97) mentions “Causal Tracing” without proper citation. I would recommend authors to cite [1].

[1] Meng et al, “Locating and Editing Factual Associations in GPT”, 2023.

**Questions For Authors:**

None

**Relation To Broader Scientific Literature:**

The paper effectively outlines key limitations in the current state of mechanistic interpretability literature, namely the limited generalization of results from task-specific circuit analysis and the high human effort required. The proposed modular circuit vocabulary approach offers a potential solution to both issues. Specifically, developing such a vocabulary could accelerate our understanding of deep neural networks by enabling researchers to leverage an existing library of modular circuits, reducing the need to repeatedly discover similar results.

**Theoretical Claims:**

None

---

> ### Author Rebuttal · Authors · 2025-04-01
>
> We sincerely appreciate your dedicated time and effort in reviewing and providing invaluable feedback. We also thank you for recognizing the novelty and the significance of our contributions. We provide a point-to-point reply below for the mentioned concerns and questions. The [anonymous repo](https://anonymous.4open.science/r/ModCirc-4887/README.md) ("the link") is used for supplementary results.
>
>
> > **Reviewer**: When constructing a significant computational subgraph (SCS) of a task, ModCirc's computation node filtration criterion may cause the SCS to be as large as 320, so explore alternative node selection methods. For instance,  retain the top N nodes necessary to achieve a faithfulness score of 1.
>
> **Answer**: We apologize for any confusion. Our node selection process identifies the top ten nodes for each of the four node types (attention head, o_projection, mlp_in, and mlp_out) at each layer. This results in approximately 320*4=1280 nodes in total. We follow your suggestion to apply faithfulness as an alternative node selection criterion, it produces slightly more nodes: approximately 2400-2500 nodes across different tasks (Symptom2Disease: 2488, Medal: 2493, MedMcqa: 2496, MedicalAbstract: 2503). We believe both selection methods can produce suitable results for modular circuit generation. However, due to time constraints, we only perform experiments on finding SCSs for different datasets on MedLLaMA, but do not derive modular circuits from the faithfulness-based selection.
>
> > **Reviewer**: Conduct causal intervention to test if the generated functional interpretations are correct.
>
> **Answer**:  The functional interpretations of the modular circuits generated by our ModCirc method can faithfully reveal its functionality. Here, we show some example results for modular circuit 18 in MedLLaMA through causal intervention (see Fig. 3 and 4 in the link). Our ModCirc identifies this modular circuit as performing "dosage detection." To verify this with causal intervention, we create a test dataset. In the dataset, each data item is a sentence containing at least one word related to dosage. We corrupt the dataset in two ways: (1) masking dosage tokens and (2) masking random non-dosage tokens. Results in Fig.3 and Fig. 4 show that masking dosage tokens significantly reduces circuit 18 activation, while random token masking either maintains or increases activation levels. The increased activation with random masking may occur because shorter sequences allow the circuit to focus more effectively on dosage detection. These findings confirm our interpretation that modular circuit 18 specializes in dosage detection.
>
> > **Reviewer**: Enumerate all the modular circuits found in MedLLaMa
>
> **Answer**: Thank you for the suggestion; please see the complete modular circuits set along with their functional interpretations in "saved_results/ploted_circuits" and "saved_results/func_interp.jsonl" in the link.
>
> > **Reviewer**: How would removing the modular circuits impact the LLM's performance?
>
> **Answer**: For each discovered modular circuit, we test how its removal affects the logits of the ground-truth token on the Symptom2Disease dataset, as shown in Fig.5 in the link. Removing these circuits causes significant performance degradation, in some cases up to 60%. Interestingly, removing a few specific modular circuits increases the logits. While this performance increase is not yet well explored, we can explain why the removals caused no degradation. These particular circuits were irrelevant to the specific task (i.e., the task circuit had no intersection with those modular circuits). Note that we also have similar observations on other datasets.
>
> > **Reviewer**: Writing suggestions: 1) improve the explanation of locality; 2) include a brief conclusion for Section 4.3; 3) Cite "Causal Tracing".
>
> **Answer**: Thank you for the suggestions; we will carefully incorporate them into our work.

---

### Official Review · Reviewer_aN6A · 2025-03-15

**Overall Recommendation:** 3

**Summary:**

The authors address two key limitations in current MI research: (1) task-specificity of circuit identification and (2) high computational costs for interpreting new tasks. Their solution introduces a modular circuits (MC) vocabulary - a collection of task-agnostic functional units, each consisting of a computational subgraph with an associated functional interpretation. These modular circuits can be reused across different language tasks, enabling modular circuit interpretability while reducing costs. The core innovation is conceptualizing a task circuit as modular components shared across different language tasks rather than building separate task-specific interpretations for each new scenario.

**Claims And Evidence:**

The paper claims that "By allowing different language tasks to share common MCs, our approach enables global interpretability while reducing costs by reusing established interpretations for new tasks." However, this claim lacks sufficient experimental validation in several important ways:

- Missing Validation of Interpretation Transfer: The authors don't experimentally verify whether the functional interpretations (FIs) derived from modular circuits (MCs) actually match the ground truth interpretations when applied to new tasks. When a new task's significant computational subgraph (SCS) maps to multiple MCs from their vocabulary, there's no evaluation of how accurately the combined MC interpretations represent the true function of the new circuit.
- Assumption Without Verification: The paper assumes that the functional interpretation of a new task's SCS can be composed from the interpretations of its constituent MCs. This core assumption - that interpretations are compositional and transferable - remains untested in the presented experiments.
- Focus on Clustering Rather Than Interpretation: The evaluations primarily demonstrate the effectiveness of their clustering technique compared to baselines (basic clustering techniques) but don't measure the accuracy of the resulting interpretations when applied to new tasks.

**Essential References Not Discussed:**

Essential references are discussed.

**Experimental Designs Or Analyses:**

As mentioned earlier, the experimental design has two significant flaws:

- The paper lacks validation for its compositional interpretation assumption. It fails to compare the functional interpretations derived from combining modular components against ground truth interpretations of the original circuits, leaving the core premise of transferable interpretations untested.
- The evaluation employs problematic metrics that don't address the central claim. Rather than assessing whether the clustering produces meaningful functional interpretations, the experiments focus solely on clustering quality. The true measure of success should be whether the functional interpretation of a new task's circuit can be accurately represented by combining the interpretations of its constituent modules.

**Methods And Evaluation Criteria:**

### Lack of Validation for Interpretation Transfer and Compositionality

The paper critically fails to validate its fundamental assumption that functional interpretations (FIs) derived from modular circuits can accurately represent new tasks. While the authors claim their MC vocabulary enables reusable interpretations across different language tasks, they provide no experimental evidence comparing these composed interpretations against ground truth. The evaluation omits any verification that subgraphs can be meaningfully decomposed into modular circuits while preserving semantic interpretation. Specifically, the authors should have compared the FI of the original circuit with the FI derived from combining its modular components to verify whether their decomposition maintains interpretative fidelity. Instead of assessing the practical utility of their dictionary approach for interpreting new tasks, the paper focuses solely on the clustering quality of extracted modular circuits, leaving the core premise of transferable, compositional interpretations completely untested.

### Problematic Baseline Selection and Circular Evaluation Design
The paper's evaluation methodology employs questionable baselines (Random, Frequent Random, KMeans, and Activation-based methods) that are inherently disadvantaged against ModCirc's reinforcement learning approach. This comparison lacks rigor as ModCirc is evaluated on the same metrics it was explicitly optimized for, creating a circular evaluation framework that virtually guarantees its superior performance. More critically, these experiments fail to address the paper's central claim about functional interpretation transfer, as they demonstrate only clustering effectiveness rather than validating whether the resulting modular decomposition preserves accurate interpretations of the original circuits.

**Other Comments Or Suggestions:**

None

**Other Strengths And Weaknesses:**

As discussed above, I reiterate the key points:

Strengths:

- If the main claim about compositional functional interpretations is supported through proper evaluation (comparing the FI of original circuits to the FI derived from combined modular components), this could be a very interesting and valuable contribution to mechanistic interpretability.

Weaknesses:
1. The paper's central theoretical claim—that functional interpretations are compositional—remains unverified. The authors do not validate that a circuit's interpretation can be accurately reconstructed by combining interpretations of its modular components, nor consider that shared circuits might serve different functions when integrated into new contexts.
2. The evaluation methodology employs problematic baselines and metrics that don't address the central claim, measuring clustering effectiveness rather than whether the resulting modular decomposition preserves accurate interpretations of the original circuits.

**Questions For Authors:**

If the authors address weakness 1 (details in earlier sections), I am happy to increase the score as the contribution is potentially interesting (strength). Specifically, compare the FI of a new task circuit derived using the proposed modular approach with the FI obtained directly from the new task circuit (more details mentioned earlier).

**Relation To Broader Scientific Literature:**

This paper extends mechanistic interpretability research by proposing a modular approach to understanding LLM behavior. While prior work focused on identifying task-specific circuits, this research introduces a method for discovering reusable functional units that can be combined to interpret new tasks. The approach offers a potentially more efficient framework for understanding model behavior across diverse language tasks.

**Theoretical Claims:**

The paper's central theoretical claim—that functional interpretations of neural circuits are compositional—remains unverified. This key assumption that a circuit's interpretation can be accurately reconstructed by combining interpretations of its modular components receives no experimental validation. The authors do not consider that circuits appearing in multiple tasks might serve fundamentally different functions when combined in novel ways. Simply because modules overlap across tasks doesn't guarantee their functional roles remain consistent when integrated into new contexts.

---

> ### Author Rebuttal · Authors · 2025-04-01
>
> We sincerely thank you for the invaluable feedback. Here, we reply to the mentioned concerns and questions. The [anonymous repo](https://anonymous.4open.science/r/ModCirc-4887/README.md) ("the link") stores supplementary results.
>
> > **Reviewer**: For a new task, verify whether the functional interpretations (FIs) derived by ModCirc are transferable and composable to match the ground-truth FIs.
>
> **Answer**: We test ModCirc's transferability and composability on two new tasks whose circuits and ground-truth FIs are established in existing work: indirect object identification (IOI) [1] and acronym detection [2]. Specifically, we train ModCirc on GPT-2 Small (the LLM [1, 2] perform on) with four NLP tasks (Trec, AGNews, etc.). ModCirc generates ten modular circuits (MCs), which contain 270 computational nodes selected from over 50k nodes. The results confirm that ModCirc can generate transferable and composable FIs:
>
> **IOI task** aims to predict the indirect object (IO) in a sentence where the subject (S) appears twice. For example, Mary (IO) and Amy (S) went out, and Amy (S) handed a pen to \_\_. [1] identifies the IOI's circuit as a set of attention heads. Each of them has a name such as "induction head" served as its FI. Our findings are:
> - ModCirc covers 92% (23/25) of ground-truth circuit heads in [1];
> - **Transferability**:
>     - MC 0 identifies two Induction Heads, whose FI in [1] is to highlight prior occurred subjects. This aligns with the MC's FI of "identifying specific entities," such as nouns.
>     - MC 1 contains two Name Mover (NM) Heads that "attend to previous names [1]" and one S-Inhibition (SI) Head that "removes duplicate names identified by the NM Heads. [1]" This aligns with the MC's FI of "extracting and assessing contextual relevance," where NM Heads extract names and SI Heads assess and remove duplicates.
>     - MC 8 contains one NM Head and three Backup Name Mover (BNM) Heads. Unlike MC 1, MC 8's FI is to "create associations or pointers between entities and their attributes," which matches the NM Head's role to associate and output the indirect object.
> - **Composibility**: MC 0 detects duplicate tokens, MC 1 assesses and removes duplication, and MC 8 outputs answer, matching the circuit structure in [1].
>
> The **acronym detection** [2] predicts the acronym of a three-word phrase (e.g., Limited Liability Company -> LLC), where we cover 87.5% ground-truth circuit nodes (7/8):
> - **Consistency**: MC 0 captures Head 5.8, aligning MC FI "extracting entities" with head FI "find capital letter"; MC 1 contains letter mover (LM) heads 9.1.&11.4, aligning MC FI "assessing context" with head FI "copy capital". The "assess" here is copying procedure.
> - **Composability**: MC 0 identifies the capital letter position, and MC 1 copies the capital letter to answer position, composing circuit's FI in [2].
>
> [1] Wang, Kevin Ro, et al. "Interpretability in the Wild: a Circuit for Indirect Object Identification in GPT-2 Small." ICLR, 2023.
>
> [2] García-Carrasco, Jorge, Alejandro Maté, and Juan Carlos Trujillo. "How does gpt-2 predict acronyms? extracting and understanding a circuit via mechanistic interpretability." AISTATS, 2024.
>
>
> > **Reviewer**: ModCirc optimizes and evaluates on the same metrics, causing circular evaluations. Further, this discriminates against the baselines as they are not explicitly optimized upon those metrics—test FI transferability for comparisons to address the concern.
>
>
> **Answer**:
> - We use different metrics for evaluation and training. During evaluation we incorporate LLMs to score generated FIs for computing composability and consistency. However, we use non-LLM proxies (see Section 4.3.4) to estimate the metrics during training for efficiency and cost reasons. Thus, we do not discriminate against the baselines because the exact evaluation metrics are not used in training.
> - To avoid the concern that "ModCirc performs well because the designed metrics discriminate against baselines," we further examine the FI transferability and composability for baselines. Specifically, we conduct experiments for the baselines in the same setting as in the last response. We observe that the baselines perform poorly in transferring FIs. For example, the Random baseline can only recover one node out of 25 nodes in IOI's circuit; it does not recover any nodes from the acronyms detection's circuit, making minimal nodes that modular circuits' FIs can apply to. Besides, Random's FIs are very similar across different MCs, failing to capture various functionalities of circuits' components.
>
> **Remark**:
> - We avoid the same circuits appearing in different tasks to serve different functions by restricting tasks within a domain, such as medical language tasks (see Definition 3.1 in the paper).
> - Supplementary C.2 and C.3 are only used to display MCs, so we do not write any claims. Please see the complete circuits in "saved_results/ploted_circuits" & "saved_results/func_interp.jsonl" in the link.

---

### Decision · Program_Chairs · 2025-05-01

**Decision:**

Accept (poster)

**Comment:**

**Summary**

This paper addresses limitations in current mechanistic interpretability (MI) research by introducing the concept of modular circuits (MCs) - task-agnostic functional units within LLMs that can be reused across different language tasks. The authors formalize five criteria for evaluating modular circuit vocabularies: consistency, locality, reusability, composability, and globality. They propose ModCirc, a novel framework using reinforcement learning-based graph neural partitioning to discover modular circuits, and demonstrate its effectiveness on MedLLaMA-8B across various medical tasks. This approach enables more global interpretability while reducing the costs associated with interpreting new tasks by reusing established interpretations.

**Reasons to Accept**

- The formalization of modular circuit vocabulary and the ModCirc framework offers a potential solution to key challenges in mechanistic interpretability and could accelerate our understanding of deep neural networks (Ezqf).
- The approach extends current task-specific mechanistic interpretability patterns by enabling cross-task interpretability while reducing computational costs (4yNw).
- The paper is well-structured with clear explanations of the evaluation criteria, algorithmic steps, and experimental setups (4yNw).
- In response to reviewer concerns, the authors demonstrated that ModCirc successfully recovers ground-truth circuits with consistent and composable functional interpretations on unseen tasks (aN6A, 4yNw).

**Potential concerns and the authors' responses**

1. Reviewer aN6A raised the concern about the composability and transferability. The authors addressed this in their rebuttal with new experiments on indirect object identification and acronym detection tasks.
2. The paper relies on GPT-4-generated functional interpretations without rigorous validation by human experts (4yNw, Ezqf). The authors acknowledged this limitation due to budget constraints but demonstrated causal intervention techniques to verify interpretations.
3. The consistency scores achieved by ModCirc were nearly on par with simple "random" and "frequency" baselines, raising questions about the method's effectiveness (h4Pi). The authors explained that baselines' consistency scores appear high because they only cover small circuit segments, while ModCirc covers much more of the task circuit as shown by its higher reusability scores.
4. The ablation study showed only a minimal performance drop when the RL-based subgraph partitioning was removed (h4Pi). The authors provided additional ablation results showing more significant performance differences across datasets and demonstrated qualitative differences in the quality of the generated interpretations.